# Heat-Killed *Enterococcus faecalis* Prevents Adipogenesis and High Fat Diet-Induced Obesity by Inhibition of Lipid Accumulation through Inhibiting C/EBP-α and PPAR-γ in the Insulin Signaling Pathway

**DOI:** 10.3390/nu14061308

**Published:** 2022-03-20

**Authors:** Jin-Ho Lee, Keun-Jung Woo, Min-Ah Kim, Joonpyo Hong, Jihee Kim, Sun-Hong Kim, Kwon-Il Han, Masahiro Iwasa, Tack-Joong Kim

**Affiliations:** 1Division of Biological Science and Technology, Yonsei University, Wonju 26493, Korea; drlogos@naver.com (J.-H.L.); rmswjd@yonsei.ac.kr (K.-J.W.); mina1218@yonsei.ac.kr (M.-A.K.); hhjoonpyo@naver.com (J.H.); jihee3649@naver.com (J.K.); kihan@berm.co.kr (K.-I.H.); masa@brm.co.jp (M.I.); 2Research & Development Center, Doctor TJ Co., Ltd., Wonju 26493, Korea; auscrlt1028@gmail.com; 3Research & Development Center, Korea BeRM Co., Ltd., Wonju 26361, Korea; 4BRM Research Center, Nihon Berumu Co., Ltd., Tokyo 100-0014, Japan

**Keywords:** obesity, adipogenesis, 3T3-L1, high fat diet, *Enterococcus faecalis*, parabiotic

## Abstract

Increasing consumption of food with high caloric density and a sedentary lifestyle have influenced the increasing obesity prevalence worldwide. The recent pandemic has contributed to this problem. Obesity refers to a state in which lipid accumulates excessively in adipocytes and adipose tissues. Dried heat-killed *Enterococcus faecalis* (EF-2001) prevents allergic mechanisms, inflammation, and tumor progression. In the present study, we investigated the effects of EF-2001 on high fat diet (HFD)-induced obese rats. The degree of obesity in experimental rats was reduced after 6 weeks of oral administration of 3 mg/kg or 30 mg/kg dosages of EF-2001, indicating regulating effects in rats with HFD-induced obesity. We found that EF-2001 decreased the amounts of total cholesterol, triglyceride, and non-high density lipoprotein (HDL) in HFD-induced obese rats. The effects of EF-2001 on 3T3-L1 adipocytes stained with Oil red O stain are shown in reductions of lipid accumulation, respectively. In addition, we examined the relationships between EF-2001 treatment and mechanisms for the insulin signaling of adipogenesis in 3T3-L1 cells. EF-2001 induced down-regulation in phosphorylation of Erk, JNK, and Akt through the inhibition of insulin receptor phosphorylation. EF-2001 inhibits the expressions of C/EBP-α and PPAR-**γ**, a lipid metabolism-related transcription factor through confocal microscope observation and Western blot on 3T3-L1 adipocytes and HFD-induced obese rats. Based on our results, intake of EF-2001 significantly prevented HFD-induced obesity in rats through inhibition of C/EBP-α and PPAR-γ in the insulin signaling pathway on lipid accumulation.

## 1. Introduction

Recently, as the number of people staying at home increases worldwide, the number of obese people is increasing, making obesity a social problem around the world [1]. Obesity is medically defined as the growth of body weight, particularly white adipose tissue (WAT), which can be developed in various diseases such as hypertension, hyperlipidemia, and cardiovascular disorders [2].

Obesity refers to a state in which lipids accumulate excessively in adipocytes and adipose tissues. Obesity is related to increased cell cycle progression and size of adipocyte triglyceride gradually accumulated in other organs such as the liver and muscles, interfering with common functions and causing various diseases [3]. The adipogenesis experimental model using 3T3-L1 cells is a preadipocyte cell line that is widely used as a model for adipocyte differentiation and is widely used as part of a high fat diet (HFD) model commonly used for in vivo study of obesity [4,5,6,7]. For other obesity models, an in vitro study using 3T3-F442A, OP9, and C3H10T1/2 cells and in vivo study of HFD-induced obese OB/OB mice with leptin gene deficiency have been reported [8,9].

Adipogenic genes are derived through various channels. There are five signaling pathways of insulin, C/REB, MAPK, WNT, and TGF-β signaling related to the expression of adipogenic genes [10]. Among them, insulin signaling pathways contribute to the expression of adipogenic protein such as C/EBP-α through phosphorylation of Akt [11] and the arrest of mitotic clonal expansion (MCE), the stage of differentiation into adipocytes, is mediated by the MAPK pathway. Referring to the known role of Erk in cell proliferation, Bost et al. could reconcile these results by hypothesizing that the function of Erk in adipogenesis has to be properly controlled. The fact that the adipocyte differentiation process of 3T3-L1 requires an initial proliferative stage is supported by their hypothesis [12]. PPAR-γ has been identified in mature adipocyte P2, leptin’s protein, and is expressed in mature adipocytes or tissues. In adipocyte development and function control, the increased level of PPAR-γ is controlled [13]. In particular, excessive accumulation of lipids in visceral fat through adipogenic markers (C/EBP-α and PPAR-γ) is achieved. Additionally, adipokine secretion such as leptin, adiponectin, and TNF-α is induced. As a result, increased penetration of macrophages in adipose tissue increases the risk of degenerative diseases such as insulin resistance, diabetes, metabolic syndrome, and cardiovascular disease [14]. 

The heat-killed *Enterococcus faecalis* (EF-2001) has previously been shown to have beneficial effects on human health. including muscle atrophy prevent, anti-allergy, anti-inflammatory, anti-tumor activities, immunomodulatory, and preventive activities [15,16,17,18,19]. Additionally, EF-2001 has been studied in the context of antioxidant mechanisms [20]. However, its use might present risks such as sepsis and other systemic infections [21]. Recently, the genome sequence of EF-2001 has been revealed and that the pre-frontal local myelination via the enhancement significantly inhibits depression [22]. Recently, Fan et al. reported that EF-2001 improves the abnormal hepatic lipid mechanism in diet-induced obese (DIO) mice by reducing triglyceride (TG) accumulation and activating the AMPK signaling pathway [23]. Another study has also reported effects of EF-2001 on animal models of prostatic hyperplasia [24]. However, the biological effect and mechanism of heat-killed parabiotic EF-2001 on HFD-induced obese rats and adipogenesis in 3T3-L1 cells have yet to be clarified.

In this study, we examined the effects of EF-2001 on HFD-induced obese rats. In addition, we conducted research on the inhibition of lipid accumulation in adipocyte and adipose tissue due to treatment with EF-2001, a heat-killed parabiotic, and examined the relationships between EF-2001 treatment and mechanisms for the insulin signaling of adipogenesis in 3T3-L1 cells 

## 2. Materials and Methods

### 2.1. Preparation of Enterococcus faecalis 2001 

EF-2001 originated from human feces is a merchantable quality parabiotic purified from Korea BeRM Co., Ltd. (Wonju, Korea) and is supplied as a heat-killed, dried powder. Dried EF-2001 included 7.5 × 10^12^ units per gram prior to being heat-killed [15].

### 2.2. Animal Experiments 

We purchased 24 male Sprague–Dawley rats (3 weeks old) with an initial body weight of 30–40 g (Orient Bio Tech Laboratories, Gyeonggi, Korea). The rats were acclimatized for one week and food intake measurements were started. The rats were categorized into 4 groups (*n* = 6/group) by diet. While there were experimental procedures in HFD-induced obese rats, the standard diet (SD) group was provided 5L79 (Lab Diet Inc., St. Louis, MO, USA). High fat diet (HFD) groups were provided with D12492 (Research Diets, Inc., New Brunswick, NJ, USA) for 6 weeks. They were subcategorized into three groups as only water, 3 mg/kg EF-2001, and 30 mg/kg EF-2001 in water. Water and food were freely available at all times. Rats in the HFD group were orally administered with pure water or EF-2001 (3 mg/kg or 30 mg/kg) in water once per day. Gavage was continued for 6 weeks. All experimental procedures were confirmed by the Institutional Animal Care and Use Committee of Yonsei University and performed according to approved guidelines (YWCI-202102-003-01).

### 2.3. Serological Analysis

Blood serum were sampled at 6 weeks by heart puncture under ether anesthesia using sterilized vacutainer tubes. Serum samples were analyzed in terms of the lipid profile, including levels of total cholesterol, triglycerides (TG), high-density lipoprotein (HDL), and non-high-density lipoprotein (non-HDL). Each lipid profile kit for total cholesterol, TG, HDL, and non-HDL detection was purchased from Asan Pharmaceutical (Asan Pharmaceutical, Seoul, Korea) and used following the manufacturer’s protocol.

### 2.4. 3T3-L1 Cell Culture and Differentiation 

The 3T3-L1 cells were routinely cultured in growth medium (GM) consisting of DMEM supplemented with 10% bovine serum (BS) (Gibco, St. Brooklyn, NY, USA) and 1% penicillin/streptomycin. The day that 3T3-L1 preadipocytes were seeded was designated as Day 2. Differentiation of 3T3-L1 preadipocytes were induced on Day 0 for 48 h in a differentiation medium (DM), consisting of DMEM supplemented with 10% fetal bovine serum (FBS) and 1% penicillin/streptomycin, mixed with MDI cocktail. MDI cocktail was made up of 500 μM of isobutylmethylxanthine (IBMX), 10 μM of dexamethasone, and 10 μg/mL of insulin. Each well was replaced to DM treated with 10 μg/mL of insulin on Day 2 and Day 4. The above-mentioned adipogenesis protocol on 3T3-L1 cells were proceeded from Day 2 to Day 6.

### 2.5. Oil Red O Staining of 3T3-L1 Adipocyte 

ADIPOGENESIS was confirmed by Oil red O (ORO) staining. EF-2001 was treated with differentiation induction medium at doses of 0, 25, 50, 100, and 250 μg/mL at 100% cell confluence, and the protocol for inducing adipocyte differentiation was performed. 3T3-L1 cells were washed with phosphate-buffered saline (PBS) and fixed with 3.7% formaldehyde (Junsei Chemical, Tokyo, Japan) diluted in PBS and stained with 60% ORO diluted in distilled water. Quantification of lipid accumulation was collected by eluting the stain in 100% isopropanol and subjecting it to microplate measurement (Molecular Devices, San Jose, CA, USA) at 490 nm. Results were represented in a graph. The percentage of ORO stain gained was relative to untreated control cells, representing the percentage of stained intracellular lipid droplet.

### 2.6. Cell Cycle Analysis 

To analyze the effects of EF-2001 on cell cycle progression, 100% confluent preadipocytes were treated with an adipogenic dexamethasone and insulin (MDI) cocktail in the absence or presence of 25, 50, 100, or 250 μg/mL of EF-2001 for 48 h. Cells were treated with 70% ethyl alcohol for over 2 h at 4 °C. EF-2001 was observed to inhibit adipogenic differentiation, and then samples were washed twice with PBS after staining with 1 mL of propidium iodide (PI) solution containing 100 μg/mL PI with PBS, 20 μg/mL of RNase A, and 0.1% NP40. G0/G1, S, and G2/M phases of cell cycles were measured using a fluorescence-activated cell sorting (FACS) system (BD Bioscience, San Jose, CA, USA).

### 2.7. Western Blotting Analysis

The 3T3-L1 cells and adipose tissues of HFD-induced rats were treated with EF-2001, and each protein was added to the lysis buffer (iNtRON Biotechnology Inc., Sungnam, Korea) at the appropriate stage from Day 2 to Day 6. After sonication treatment, the protein was quantified and tested using Bradford assay (Bio-Rad, Hercules, CA, USA) for Western blotting. The SDS-polyacrylamide gel ratio was determined according to the protein kDa to be confirmed, and electrophoresis was performed at 100 V for approximately 2 h. Antibody treatment was performed with primary antibodies (C/EBP-α, PPAR-γ, p-Erk, Erk, p-JNK, JNK, p-Akt, Akt, p-IR, IR, and β-actin) at a rate of 1:2500 overnight at 4 °C. It was washed three times with Tris-buffered saline solution containing Tween 20 for 10 min, and then secondary antibodies were added at a rate of 1:5000 for 2 h at room temperature (RT). The transferred protein band of the PVDF membrane was measured using the LAS 4000 system (GE Healthcare, Little Chalfont, UK) by inducing an enhanced chemiluminescence reaction. 

### 2.8. Confocal Microscopy

For confocal microscopy, 3T3-L1 preadipocytes were cultured in a 3 cm plate of cover glass (Mattek Corp, Ashland, MA, USA), differentiated, and treated with each dose of EF-2001. To facilitate observation of the nucleus, cells were cultured for 10 min by immobilizing paraformaldehyde at RT with fluorescent dye DAPI diluted in PBS. To visualize C/EBP-α and PPAR-γ antibody, the cells were incubated with fluorescent dye GFP diluted in PBS for 30 min at RT with paraformaldehyde fixation. GFP expression was visualized with a LSM710 confocal microscope (Carl Zeiss, Oberkochen, Germany).

### 2.9. Statistical Analysis

All values are expressed as mean ± SEM. The data were analyzed by one-way ANOVA (Student’s t-test) using PRISM version 5.0. The differences among groups were assessed using Dunnett’s test. Statistical significance was indicated by *p*-values ** p* < 0.05, *** p* < 0.01, and *****
*p* < 0.001.

## 3. Results

### 3.1. EF-2001 Intake Effectively Decreases White Adipose Tissue and Body Weights on HFD-Induced Obese Rats

To establish an HFD-induced obesity model, male rats were divided into SD or HFD. Rats were orally administered with refined water or EF-2001 in water for each dose per day as scheduled (Figure 1A). HFD groups were subcategorized into three groups (only refined water, 3 mg/kg, or 30 mg/kg EF-2001 in water) to perform the effects of EF-2001 treated on obese rats. We investigated the effects of EF-2001 intake on HFD-induced elevation in body weight. Rats fed with HFD weighed significantly more than rats fed with SD. Both groups of HFD rats administered with 3 mg/kg or 30 mg/kg of EF-2001 resulted in significant weight reduction (Figure 1B). White adipose tissue weight per body weight was higher in HFD-fed rats than in SD-fed rats and EF-2001 significantly reduced HFD-induced white adipose tissue (Figure 1C,D). 

### 3.2. EF-2001 Down-Regulates the Levels of Total Cholesterol, TG, and Non-HDL in Serum of High Fat Diet-Induced Obese Rats

The effects of EF-2001 on the level of serum lipid profiles in HFD-induced obese rats are shown in Figure 2. HFD-induced obese rats supplemented with EF-2001 (3 mg/kg or 30 mg/kg dose group) for 6 weeks showed a remarkable reduction in the levels of total cholesterol (Figure 2A), TG (Figure 2B), and non-HDL (Figure 2D) than HFD-induced obese group without EF-2001. In this study, no significant difference was observed in HDL levels between the four groups. (Figure 2C).

### 3.3. EF-2001 Inhibits Lipid Accumulation in Differentiated 3T3-L1 Adipocytes

Inhibition effect of EF-2001 on lipid accumulation is presented as ORO-stained 3T3-L1 adipocytes. Data are shown in Figure 3A. Treatment with EF-2001 decreased the level of ORO staining in a dose-dependent manner. Additionally, EF-2001 inhibited the accumulation of intracellular lipids. DM with 100 or 250 μg/mL of EF-2001 resulted in 15 and 30% reduction in lipid accumulation compared with the untreated control (Figure 3B). 

### 3.4. EF-2001 Delays MDI-Induced Cell Cycle Progression in 3T3-L1 Adipocytes

Differentiated 3T3-L1 adipocyte with EF-2001 treatment were analyzed by flow cytometry to examine mitotic clonal expansion (MCE) [25]. When treated with various doses of EF-2001 such as 25, 50, 100, and 250 μg/mL for 48 h, the percentage of S phase increased in EF-2001 treatment groups compared with the control group, while the percentage of the G2/M phase in the EF-2001 treatment group decreased in a dose-dependent manner. Therefore, after EF-2001 treatment, cell cycle progression was arrested in G0/G1phase. (Figure 4). 

### 3.5. EF-2001 Inhibits MDI-Induced Insulin Receptor Pathways in 3T3-L1 Adipocytes

IBMX and MDI are important initiators of cell cycle progression, contributing to the initiation of MCEs during adipogenesis. In addition, insulin treatment is important for the growth of lipid-droplets in differentiated adipocytes after MCE [26]. Therefore, we examined whether EF-2001 affects insulin signaling pathways. The receptor sub-signal significantly inhibited the MDI-induced phosphorylation of Erk, JNK, and Akt as the dose of EF-2001 increased (Figure 5A–C). Insulin stimulation induced auto-phosphorylation of IR-β subunit in tyrosine residues. Treatment with EF-2001 had an inhibitory effect on the phosphorylation level of IR (Figure 5D).

### 3.6. EF-2001 Inhibits the Protein Expression and Nuclar Translocation of C/EBP-α and PPAR-γ in 3T3-L1 Adipocytes and in HFD-Induced Adipose Tissues

During 3T3-L1 adipogenesis, C/EBP-α and PPAR-γ are activated by insulin with FBS in DM [27,28]. In addition, the experiments were conducted during adipogenesis to investigate how EF-2001 treatment regulates the expression level of transcription factors such as C/EBP-α and PPAR-γ. The protein expressions of C/EBP-α was meaningfully inhibited by EF-2001 (Figure 6A). Even in confocal microscopy, C/EBP-α, identified as Alexa-488, significantly decreased protein expression in the nucleus (Figure 6B). The protein expression of PPAR-γ was significantly decreased with EF-2001 treatment (Figure 6C). In addition, the PPAR-γ protein expression in the nucleus was remarkably decreased in 250 μg/mL EF-2001-treated cells (Figure 6D). Eventually, we investigated the effect of EF-2001 on the protein expression level of C/EBP-α and PPAR-γ in adipose tissues of SD-fed rats and HFD-induced rats. Analyses of protein samples extracted from white adipose tissue were confirmed by Western blot. Oral administration of EF-2001 (30 mg/kg) on the HFD group effectively decreased the protein expression level of C/EBP-α to a lower level than the untreated EF-2001 HFD group (Figure 7A,B). Oral administration of EF-2001 (3 mg/kg or 30 mg/kg) on the HFD group effectively decreased the protein expression level of PPAR-γ to a lower level than the untreated EF-2001 HFD group (Figure 7A,C).

## 4. Discussion

In this study, an obese rat model was induced by feeding an HFD. The degree of obesity in experimental rats was reduced after 6 weeks of oral treatment with 3 mg/kg and 30 mg/kg dosages of EF-2001, indicating regulating effects in rats with HFD-induced obesity. Additionally, increasing adipogenesis is a common mechanism underlying obesity [29,30]. We found EF-2001 to significantly reduce adipose tissue accumulation (Figure 1). These results support that orally administrated EF-2001 effectively decreases HFD-induced accumulation of white adipose tissue and obesity. 

Hyperlipidemia is strongly connected with obesity and cardiovascular disease [31]. We demonstrated that oral administration of EF-2001 benefited lipid metabolism, including cholesterol-lowering. The results showed that EF-2001 significantly lowered plasma TG level at both doses (3 mg/kg and 30 mg/kg EF-2001). In addition, both doses significantly reduced non-HDL cholesterol levels in HFD-induced rats but does not affect HDL (Figure 2). An HFD can increase total cholesterol (such as TG and LDL-cholesterol levels) resulting in an increased risk of atherosclerosis [32]. LDL cholesterol is known to be the primary factor that induces atherosclerotic plaque lesions by contributing to the penetration of oxidized LDL cholesterol into arterial walls. Regulating serum TC and LDL cholesterol level is required to prevent the formation of atherosclerotic plaque lesions [33]. These results show that EF-2001 contributes to the down-regulated control of levels of serum TG, total cholesterol, and non-HDL cholesterol elevated by the induction of HFD.

The adipogenesis experimental model using 3T3-L1 cells is a preadipocyte that is one of the models widely used for adipogenesis study [4,5,6,7]. Other groups report that products that utilize bacteria such as *Lactobacillus sakei ADM14, L. brevis OPK-3,* and *L. plantarum LMT1-48* inhibit the accumulation of lipid droplets [34,35,36,37]. The effects of EF-2001 on 3T3-L1 adipocytes stained with ORO are examined by reduction of lipid accumulation (Figure 3). Activation of insulin receptors and MDI treatment induce the differentiation of pre-adipocytes into adipocytes. It is a crucial function in MCE of early differentiation induction [38,39]. Our data showed an increase in S phase and a decrease in G2/M phase by EF-2001 (Figure 4). Therefore, we suggest that EF-2001 induces the inhibition of early cell cycle progression. In our experiments, proteins that regulate each stage of the cell cycle, such as cell cycle-dependent kinases (CDKs), have not been identified. The exact identification and mechanism of cell cycle regulatory proteins in the current HFD model require further study.

Insulin signaling pathways are involved in intracellular lipid production by insulin in 3T3-L1 cells [40]. Insulin receptor pathways and phosphorylation of MAPKs are affected by insulin [41]. We analyzed the molecular mechanisms by which EF-2001 inhibits the adipogenic process in 3T3-L1 cells to confirm the anti-lipid accumulation of EF-2001. EF-2001 inhibited phosphorylation of insulin signaling pathway proteins such as IR, Erk, Akt, and JNK signals that mediate adipogenesis (Figure 5). These results show that EF-2001 inhibits insulin signaling pathways, causing a cascade of phosphorylation beginning with IRs and proceeding to downstream signals such as Erk, JNK, and Akt.

Additionally, we confirmed that the insulin signaling pathway induced C/EBP-α and PPAR-γ known for the adipogenic transcription factors. Other studies have noted the importance of regulating adipogenic transcription factors in the regulation of adipogenesis and also related to adipogenesis modulators such as aP2 and FAS that initiated and regulated adipogenesis in 3T3-L1 adipocytes [42,43].

Consequently, EF-2001 inhibits the expression of C/EBP-α and PPAR-γ, lipid metabolism-related transcription factors in 3T3-L1 adipocytes and HFD-induced obese rats (Figure 6 and Figure 7). A recent obesity study reports that the fractionation products or extracts derived from *Staphylococcus* and *Lactobacillus* inhibit triglyceride accumulation through changes in C/EBP-α and PPAR-γ in 3T3-L1 adipocyte differentiation [44].

We identified lipid droplet reduction by EF-2001 treatment in 3T3-L1 adipocyte. This is related to the inhibition of PPAR-γ and C/EBP-α which are the key transcription factors of adipogenesis. Recent studies also reported a relationship between the insulin signaling pathway and C/EBP-β and adipogenic transcription factor-induced adipogenesis [45]. Furthermore, inhibition of C/EBP-α and PPAR-γ reduces lipid accumulation in WAT and can therefore contribute to obesity prevention. Additionally, it can reduce the risk of diseases such as obesity-induced metabolic syndrome and vascular diseases [46,47]. Therefore, our results suggest that EF-2001 inhibited adipogenesis by reduction of C/EBP-α and PPAR-γ expression and nuclear translocation both in vitro and in vivo.

Recently, another notable study has reported the influence of weight control based on the impact of EF-2001 administration on the AMPK pathway in mice with obesity induced by HFD. Fan et al. reported that EF-2001 remarkably elevated the expressions of p-AMPK and p-ACC in mice and suggested that EF-2001 decreases hepatic lipid accumulation in the DIO model mice via AMPK pathway while improving livers injured by an HFD [23]. AMPK is an important factor in regulating homeostasis and basal metabolism in mammalian cells [48]. The AMPK is a protein marker of lipolysis which demonstrates the effects of drugs that induce lipid production in vivo or in vitro and activates lipolysis enzymes (such as HSL, ATGL, and MGL). Phosphorylation of AMPK mediates mechanisms related to beta oxidation or ATP consumption [49]. However, it is necessary to analyze the signaling mechanisms directly related to lipid metabolism in vivo, in adipocytes, and tissues, and to study the mechanisms of EF-2001 from differentiation induction of adipocytes to lipolysis, which is already produced and induced or repeated production and decomposition. Therefore, there is a difference in the order of intracellular signaling resulting from treatment with EF-2001 during adipogenic induction. For a clear relevance analysis, this aspect will require further study.

Dosage determination of EF-2001 was calculated based on the dose administered to humans. The adult clinical dosage of EF-2001 is 1.5 g per session and once a day. Converting this dose to an adult weight of 60 kg, the clinical dosage for adults is 25 mg/kg/day. The equivalent dose for mice is 12.3 times that of human adults, 307.5 mg/kg/day probiotics [23]. Additionally, according to previous reports on EF-2001, various oral concentrations of EF-2001 were administered in our mouse experiments. In Fan’s group, EF-2001 was administered to mice once a day at 200 mg/kg [23]. In Takahashi’s group, 250 mg/kg of EF-2001 was used in mice [50]. However, the results of toxicity tests with oral administration have been recently reported. It was estimated that 50% lethal dose upon oral administration of EF-2001 to mice would be more than 5000 mg/kg body weight/day for both male and female mice [51]. We did not conduct a toxicity test of EF-2001 in SD rat experiments, but the maximum concentration administered was 30 mg/kg. This concentration is much lower than those used in other groups and is safe based on the results of toxicity tests.

The use of heat-killed parabiotic EF-2001 has a functionally similar effect on probiotics along with potential risks. Heat-killed parabiotics have economic benefits such as extending the shelf life of products and convenient transportation and storage. Our study could contribute to the reductions of various diseases, including insulin resistance, in obese people due to decreases in adipokine-secretive adipocytes. In order to clarify the effects of parabiotics on other lipogenesis and lipolysis mechanism regulators, the components of EF-2001 that can contribute to inhibiting adipogenesis could be analyzed and identified through further research.

## 5. Conclusions

In this study, we identified the effects and mechanisms of EF-2001 in the adipogenesis process and obesity. EF-2001 showed regulating effects on HFD-induced obesity, and decreased the amounts of total cholesterol, triglycerides, and non-HDL in HFD-induced obese rats. EF-2001 induced down-regulation in phosphorylation of downstream signals such as Akt, Erk, and JNK through insulin receptor signaling pathways. Furthermore, oral administration of EF-2001 significantly prevented HFD-induced obesity in rats and reduced protein expression levels of C/EBP-α and PPAR-γ in adipose tissue (Figure 8). Based on our results, the insulin signaling pathway is a potential target of EF-2001 in the adipogenesis process. 

## Figures and Tables

**Figure 1 nutrients-14-01308-f001:**
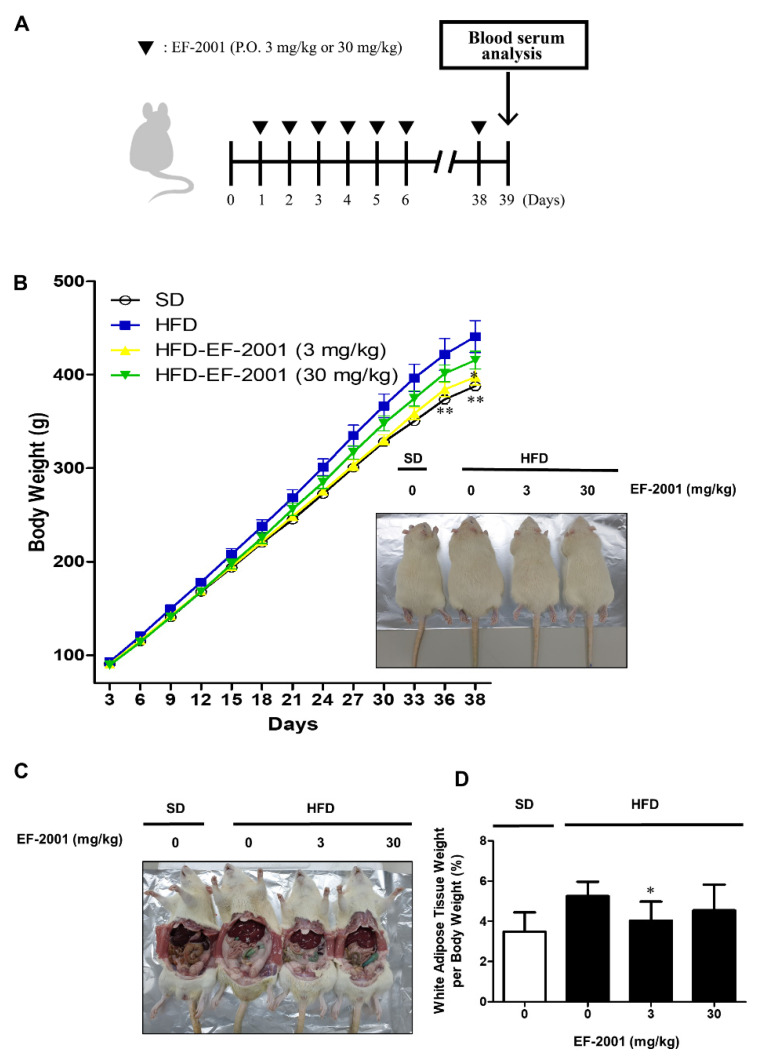
Effects of EF-2001 oral administration on HFD-induced accumulation of white adipose tissue and body weights of obese rats. (**A**) Workflow of experimental procedure in HFD-induced SD rats. Male rats were divided into standard diet (SD) or high fat diet (HFD), and rats were orally administered with refined water, 3 mg/kg EF-2001, or 30 mg/kg EF-2001 in water (*n* = 6 rats/group). Gavage was continued for 6 weeks. (**B**) Effects of EF-2001 on the weight of SD rats fed by standard diet and HFD for 6 weeks. Body weight was monitored each day. (**C**) Effects of EF-2001 on the accumulation of adipose tissue of HFD-induced obese rats. To investigate the weight of adipose tissue, the rats were sacrificed, and the abdomen was opened to take pictures. (**D**) Effects of EF-2001 on the weight of white adipose tissue of HFD-induced obese rats. Data was analyzed with white adipose tissue weight per body weight. Data are represented as the mean ± SEM. (*n* = 6); * *p* < 0.05 and ** *p* < 0.01 vs. control (without EF-2001 in HFD-induced obese rats).

**Figure 2 nutrients-14-01308-f002:**
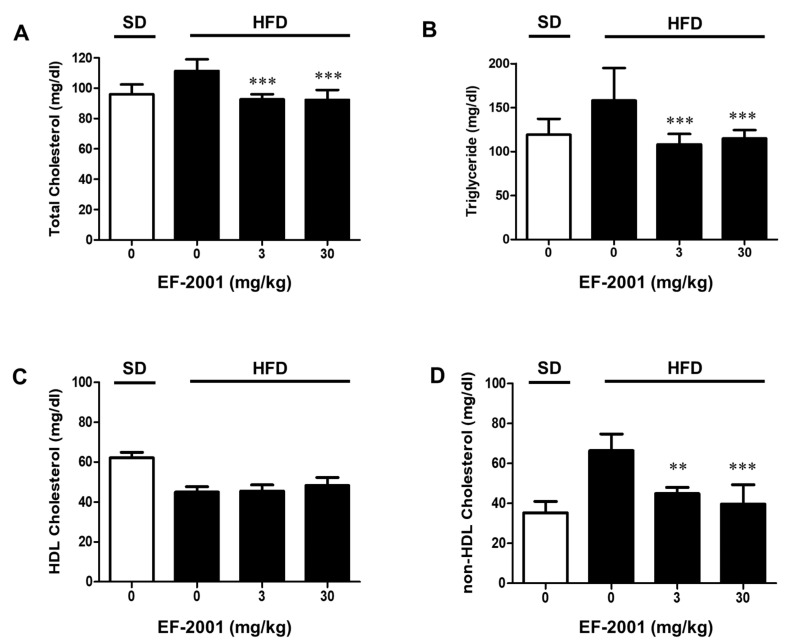
Effects of EF-2001 on the levels of total cholesterol, TG, HDL, and non-HDL in serum of high fat diet-induced obese rats. After six weeks of feeding, rats were fasted for 12 h, and then blood was collected. The effects of EF-2001 on the contents of total cholesterol (**A**), triglycerides (**B**), HDL (**C**), and non-HDL levels (**D**) in HFD-induced serum were measured. Data are represented as the mean ± SEM. (*n* = 6); ** *p* < 0.01, *** *p* < 0.001 vs. control (without EF-2001 in HFD-induced obese rats).

**Figure 3 nutrients-14-01308-f003:**
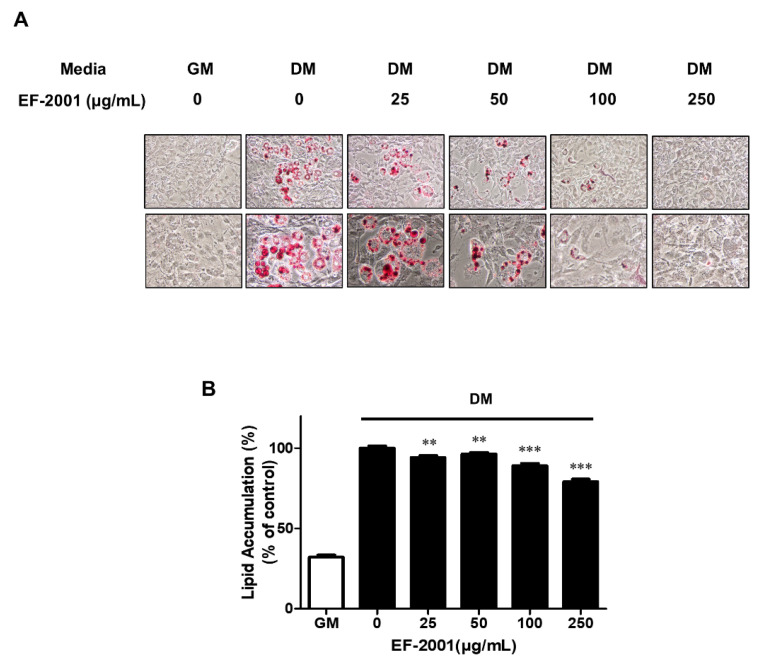
Effects of EF-2001 on lipid accumulation of 3T3-L1 adipocyte differentiation. (**A**) Effects of EF-2001 on differentiated 3T3-L1 adipocyte with ORO staining. 3T3-L1 preadipocytes were cultured with growth medium (GM) or differentiation medium (DM) in the absence or presence of EF-2001 (25–250 μg/mL) for 6 days. Lipid accumulation of 3T3-L1 adipocyte was observed by ORO staining. (**B**) Effects of EF-2001 on the relative lipid accumulation of 3T3-L1 adipocyte differentiation. ORO-stained 3T3-L1 cells were eluted with 100% isopropanol, and optical density (O.D.) was measured at 490 nm. Data are represented as mean ± SEM. (*n* = 4); ** *p* < 0.01 and *** *p* < 0.001 vs. control (without EF-2001 in differentiation medium).

**Figure 4 nutrients-14-01308-f004:**
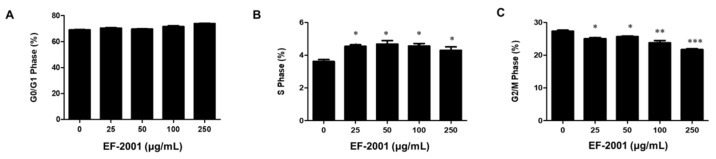
Effects of EF-2001 on MDI-induced cell cycle progression in 3T3-L1 adipocytes. MDI-treated 3T3-L1 preadipocytes were cultured with DM in the presence or absence of EF-2001. Cells treated with MDI for 48 h were stained with propidium iodine (PI) and analyzed by flow cytometry. The percentages of the cell population at G0/G1 (**A**), S (**B**), and G2/M (**C**) phase of the cell cycle were calculated from the data. Data are represented as mean ± SEM. (*n* = 4); * *p* < 0.05, ** *p* < 0.01, and *** *p* < 0.001 vs. control (without EF-2001 in differentiation medium).

**Figure 5 nutrients-14-01308-f005:**
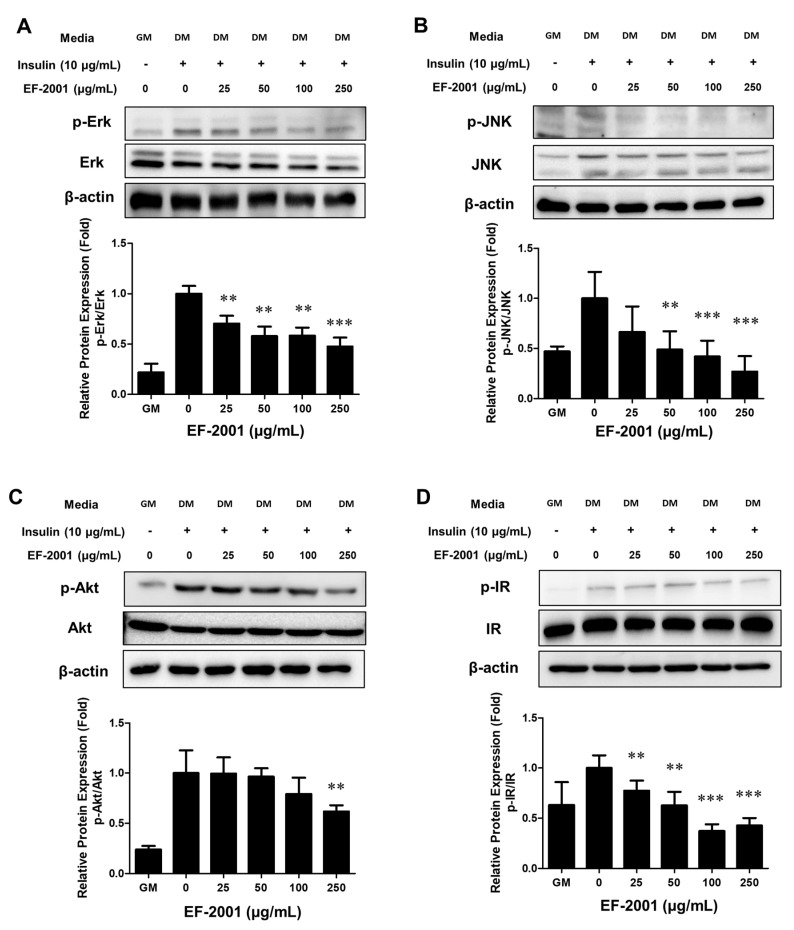
Effects of EF-2001 on insulin receptor pathways in 3T3-L1 adipocytes. 3T3-L1 adipocytes were performed to adipogenesis for 2 days in the absence or presence of various dose of EF-2001. Cells were harvested at the time at which the expression of each insulin signaling protein was induced. Cells were sampled by lysis buffer after insulin treatment for phosphorylated Erk (30 min) (**A**), phosphorylated JNK (45 min) (**B**), phosphorylated Akt (45 min) (**C**), and phosphorylated IR based on phosphorylation detection (15 min) (**D**). Data are represented as mean ± SEM. (*n* = 3); ** *p* < 0.01, and *** *p* < 0.001 vs. control (without EF-2001 in differentiation medium).

**Figure 6 nutrients-14-01308-f006:**
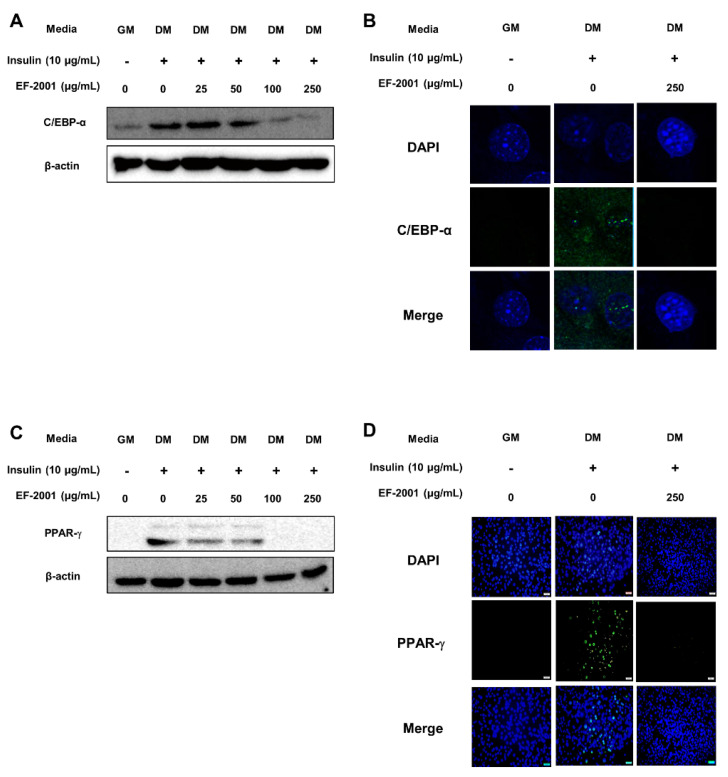
Effects of EF-2001 on the protein expression and nuclear translocation of C/EBP-α and PPAR-γ in 3T3-L1 adipocytes. 3T3-L1 adipocytes were cultured in the absence or presence of EF-2001 at various dose (0, 25, 50, 100, and 250 μg/mL) on MCE and insulin treated after the MCE stage for each medium condition. (**A**) Effect of EF-2001 on the protein level of CCAAT/enhancer binding protein (C/EBP)-α in MDI-induced 3T3-L1 adipocytes. Lysate for C/EBP-α was sampled on Day 4. The C/EBP-α immunoblot was represented by three independent replicates. (**B**) Effect of EF-2001 on the nuclear translocation of C/EBP-α in MDI-induced 3T3-L1 adipocytes. C/EBP-α was marked by anti-C/EBP-α primary antibody with Alexa 488 (green), and nuclei were stained with 4′6-diamidino-2-phenylindole (DAPI) (blue). The translocation of C/EBP-α was represented with confocal microscopy at 400× magnification. (**C**) Effect of EF-2001 on the protein level of peroxisome proliferator-activated receptor (PPAR)-γ in MDI-induced 3T3-L1 adipocytes. Lysate for PPAR-γ was sampled on Day 6. The PPAR-γ immunoblot was represented by three independent replicates. (**D**) Effect of EF-2001 on the nuclear translocation of PPAR-γ in MDI-induced 3T3-L1 adipocytes. PPAR-γ was marked by anti-PPAR-γ primary antibody with Alexa 488 (green), and nuclei were stained with DAPI (blue). The translocation of PPAR-γ was represented with fluorescence microscopy at 100× magnification.

**Figure 7 nutrients-14-01308-f007:**
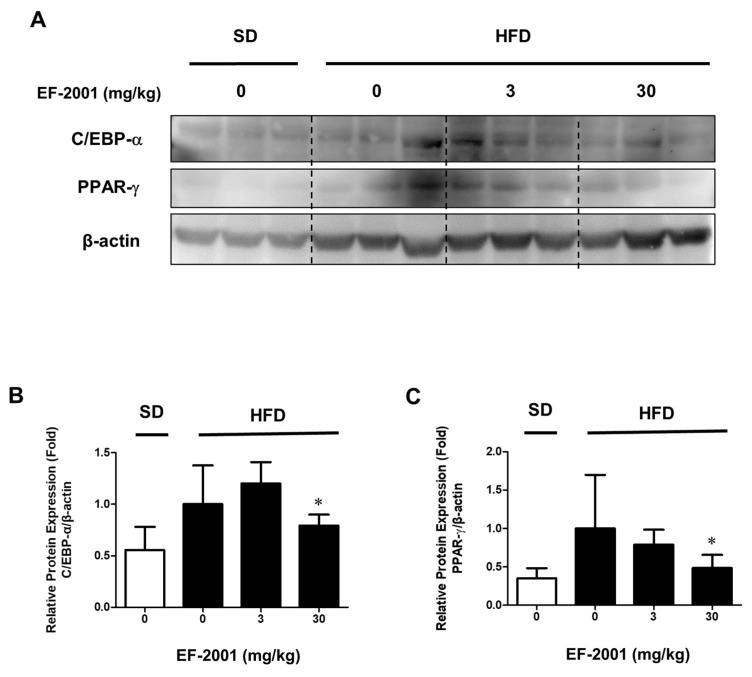
Effects of EF-2001 on immunoblot for the protein level of C/EBP-α and PPAR-γ in HFD-induced adipose tissues. Male rats were divided into SD or HFD, and rats were orally administered with refined water, 3 mg/kg, or 30 mg/kg EF-2001 for 6 weeks (*n* = 6 rats/group). White adipose tissues containing epididymal fat were collected and separated into lipid and protein. C/EBP-α and PPAR-γ were detected by immunoblotting. (**A**) The protein levels of C/EBP-α and PPAR-γ were standardized depending on the amount of β-actin. The relative protein expression levels of C/EBP-α (**B**) and PPAR-γ (**C**) are shown in the graphs. Data are represented as the mean ± SEM. (*n* = 6); * *p* < 0.05 vs. control (without EF-2001 in HFD-induced obese rats).

**Figure 8 nutrients-14-01308-f008:**
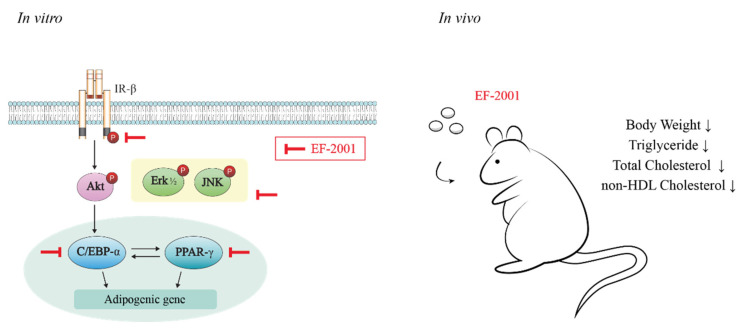
Schematic diagram of anti-obesity activity of EF-2001 on 3T3-L1 adipocytes and HFD-induced obese rats.

## Data Availability

The data that support the findings of this study are available from the corresponding author upon reasonable request.

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
