# Peer review of "Heat-Killed Enterococcus faecalis Prevents Adipogenesis and High Fat Diet-Induced Obesity by Inhibition of Lipid Accumulation through Inhibiting C/EBP-α and PPAR-γ in the Insulin Signaling Pathway"

_nutrients, 2022, doi:10.3390/nu14061308_

Round 1

Reviewer 1 Report

In this study, Lee et al. evaluated the anti-lipogenic effect of Enterococcus faecalis in vivo and in cell cultures. The results are interesting, however, there are a number of points that need to be addressed.

- It does not seem appropriate to this reviewer to focus the reason for the increase in obesity on covid-19, there are many other factors that had already triggered its prevalence before the pandemic. Please review this point again in the abstract. You can say that in addition to the high consumption of foods with high caloric density, the little activity caused by a sedentary life, which has been increased by the recent pandemic.

- This reviewer is struck by the fact that the authors refer that the effect of Enterococcus faecalis in HFD is unclear, despite the fact that a recent study reported (doi: 10.3390/foods11040575) the same anti-lipogenic effect in this model, through the of AMPK. Even less understandable is the fact that the introduction does not cite this work.

- In the abstract, harmonize your conclusion with the title of the manuscript.

- Authors should point out the rationale for EF-2001 dose selection.

- The authors do not indicate that the blood was collected under anesthesia; if the animals were not anesthetized, it is very likely that the measured parameters were altered by the stress generated by the cardiac puncture.

- The methodology would be greatly benefited if it were supported by citing previous works using the same protocols.

- It is not clear how the EF-2001 was administered, was it by gavage?

- Adipose tissue (Fig. 1C) should be normalized to weight.

- In section 2.7 the authors talk about analysis done on mice, however the study used rats. Also line 199 and 318, etc... Please clarify that.

- Change ml to mL on the line 117, 217, etc..

Author Response

Response to Reviewer 1 Comments

- It does not seem appropriate to this reviewer to focus the reason for the increase in obesity on covid-19, there are many other factors that had already triggered its prevalence before the pandemic. Please review this point again in the abstract. You can say that in addition to the high consumption of foods with high caloric density, the little activity caused by a sedentary life, which has been increased by the recent pandemic.

Response to Reviewer Comment : We agree with you. The Covid-19 pandemic has been affecting the whole world and its activity levels. Therefore, we included this in the abstract as a cause of obesity. We have modified the sentence according to your comment

- This reviewer is struck by the fact that the authors refer that the effect of Enterococcus faecalis in HFD is unclear, despite the fact that a recent study reported (doi: 10.3390/foods11040575) the same anti-lipogenic effect in this model, through the of AMPK. Even less understandable is the fact that the introduction does not cite this work.

Response to Reviewer Comment : Thank you for pointing this out. The suggested paper (doi: 10.3390/foods11040575) was published online on February 16, 2022, the same day that our study was released. Because we could not confirm its contents, we did not include it in our paper. Based on your comment, we incorporated findings from that study in the Abstract, Introduction, and Discussion of our revised manuscript.

- In the abstract, harmonize your conclusion with the title of the manuscript.

Response to Reviewer Comment : Based on your comments, we have matched the title and abstract.

- Authors should point out the rationale for EF-2001 dose selection.

Response to Reviewer Comment : Dosage determination of EF-2001 was calculated based on the dose administered to humans. The adult clinical dosage of EF-2001 is 1.5 g, once a day. After conversion, the clinical dosage for adults is 25 mg/kg/day (adult weight is considered 60 kg). The equivalent dose for mice is 12.3 times that of human adults [Reference 1], 307.5 mg/kg/day probiotics.

Also, according to previous reports on EF-2001, various oral concentrations of EF-2001 were administered in our mouse experiments. In Kim’s group, 200 mg/kg of EF-2001 was orally administered in mice [Reference 2]. In Takahashi’s group, 250 mg/kg of EF-2001 was used in mice [Reference 3]. However, the results of toxicity tests with oral administration have been recently reported. The 50% lethal dose of EF-2001 with oral administration in mice was estimated to be greater than 5,000 mg/kg body weight/day for both male and female mice [Reference 4]. We did not conduct a toxicity test of EF-2001 in SD rat experiments, but the maximum concentration administered was 30 mg/kg. This concentration is much lower than those used in other groups and is safe based on the results of toxicity tests.

[Reference 1] Nair, A.; Morsy, M.A.; Jacob, S. Dose translation between laboratory animals and human in preclinical and clinical phases of drug development. Drug Develop. Res. 2018, 79, 373–382.

[Reference 2] Meiqi Fan, Young-Jin Choi, Nishala Erandi Wedamulla, Yujiao Tang, Kwon Il Han, Ji-Young Hwang, Eun-Kyung Kim. Heat-Killed Enterococcus faecalis EF-2001 Attenuate Lipid Accumulation in Diet-Induced Obese (DIO) Mice by Activating AMPK Signaling in Liver. Foods. 2022 Feb; 11(4): 575. 

[Reference 3] Kohei Takahashi, Osamu Nakagawasai, Wataru Nemoto, Takayo Odaira, Wakana Sakuma, Hiroshi Onogi, Hiroaki Nishijima, Ryuji Furihata, Yukio Nemoto, Hiroyuki Iwasa, Koichi Tan-No, Takeshi Tadano. Effect of Enterococcus faecalis 2001 on colitis and depressive-like behavior in dextran sulfate sodium-treated mice: involvement of the brain–gut axis. J Neuroinflammation. 2019; 16: 201. 

[Reference 4] Yeun-Hwa Gu, Takenori Yamasita, Ki-Mun Kang. Subchronic Oral Dose Toxicity Study of Enterococcus Faecalis 2001 (EF 2001) in Mice. Toxicol Res. 2018 Jan; 34(1): 55–63.

- The authors do not indicate that the blood was collected under anesthesia; if the animals were not anesthetized, it is very likely that the measured parameters were altered by the stress generated by the cardiac puncture.

Response to Reviewer Comment : We collected and analyzed blood samples under anesthesia to minimize stress in the animals. We added details on the anesthesia.

- The methodology would be greatly benefited if it were supported by citing previous works using the same protocols.

Response to Reviewer Comment : We added references that previously used the same experimental protocol.

- It is not clear how the EF-2001 was administered, was it by gavage?

Response to Reviewer Comment : The probiotics were administered directly by gavage feeding. We added a detailed description of the administration method of EF-2001.

- Adipose tissue (Fig. 1C) should be normalized to weight.

Response to Reviewer Comment : Adipose tissue weight is normalized to body weight. We revised the graph accordingly.

- In section 2.7 the authors talk about analysis done on mice, however the study used rats. Also line 199 and 318, etc... Please clarify that.

Response to Reviewer Comment : This was our mistake. We changed all such instances to rat.

- Change ml to mL on the line 117, 217, etc..

Response to Reviewer Comment : We changed ml to mL

Reviewer 2 Report

MS Nutrients-1620181

Lee J. et al investigated the effect of the functional role of microbiome products which contain het killed Enterococcus faecalis, EF-2001 on HFD fed male SD rats as an animal model of obesity.

After planning and define the objective following by relevant experiment design along with literature reviewing of a plethora of previous research, they examined to determine if EF-2001 ameliorate anti-obesity features including cell proliferation, differentiation, adipogenesis and lipid accumulation utilize molecular tool, cellular and animal study model such as 3T3_l1 preadipocyte cells and HFD fed obese rat model to unfold molecular circuit beyond the potential therapeutics.

The report is written well, adequate proof-of–evidence combination of data set justification with molecular crosstalk between anti-oxidative signaling and anti-proliferative molecular signature using cell cycle analysis and histology data using confocal observation.

However, there are some pitfalls and discrepancy between issue of EF-200-1 dosage and effectiveness along with shortage of molecular network including its cytokine regulation following EF-2001 treated HFD fed SD rat.

Please provide authors comments to the following points as below.

Is there any difference between sex or not, would you provide your criteria why choose male rat in this study?

In a previous study (Takahashi K et al (2019) used a similar product, Nihon BRMC EF-2001, is that same product or not?

I saw the author tested two different doses such as 3, and 30mg/kg EF-2001 as challenge dose, have you been tested what the toxicity level do they find. In another study, they tested 250 mg/kg of C57Bl/6 mice. How about the SD rat model, have you tested the high range of EF-2001 tested in the SD rat model?

In Fig. 4 do you have any histogram data of cell cycle? Did you observe any cell cycle dependent kinase interrupted or not? Have you tested with positive control compared to EF-2001? Do you have any animal data regarding cell proliferation data to demonstrate your hypothesis (Line 347, author mentioned EF-2001 induce inhibition of early cell cycle signals in the MCE process in both?)

Am wondering they verify the signaling crosstalk if EF-2001 activated in the rat model by confirming gene and protein level expression following EF-2001 and determine functional analysis using siRNA technology or others, what molecular targeting underlying IR-beta signaling could be activated anti-obesity effect on Scheme Fig.8.

Wondering interaction between EF-2001 and enzymatic activity in lipolysis in SD rat tissue.

Is there any evidence of AMPK signaling alteration along with cytokines /chemokines pattern following EF treatment between SD and HFD rats?

Minor issue

Would you provide chemical components with the EF-2001?

In fig 1A, there are missing figure legends regarding C. Also, do you have a HFD with EH, EL abdomen needs to be indicated when the rat is sacrificed in detail.

To clarify your experimental study, suggest a schematic workflow including treatment points of EL and EH with no subject in the animal study.

In Fig. 3 describe GM abbreviation in figure legend

Author Response

Response to Reviewer 2 Comments 

Is there any difference between sex or not, would you provide your criteria why choose male rat in this study?

Response to Reviewer Comment : We understand your opinion. However, the consistency of experimental results in female rats is lower than that of males because of the menstrual cycle and hormonal changes. This can then affect the results of data analysis. A previous study reported that a high-fat diet in adult rats resulted in less obesity and hypertension compared to such a diet in weaned male rats [Reference 1]. Therefore, many researchers have designed and conducted similar experiments using males because it takes less time and effort to impact metabolism in male mice. Our choice to use male rats also was based on their use in previous studies.

[Reference 1] Hong Sheng Cheng, So Ha Ton, Sonia Chew Wen Phang, Joash Ban Lee Tan, Khalid Abdul Kadir, Increased susceptibility of post-weaning rats on high-fat diet to metabolic syndrome, Journal of Advanced Research, 2017 Nov; 8(6) 743-752

In a previous study (Takahashi K et al (2019) used a similar product, Nihon BRMC EF-2001, is that same product or not?

Response to Reviewer Comment : These are the same product. EF-2001 has production plants in Japan and Korea. The BRMC EF-2001 by Takahashi K et al (2019) was produced in Japan [Reference 1]. The EF-2001 we used was produced in Korea using the same cell line cultured and killed as in Japan.

[Reference 1] Kohei Takahashi, Osamu Nakagawasai, Wataru Nemoto, Takayo Odaira, Wakana Sakuma, Hiroshi Onogi, Hiroaki Nishijima, Ryuji Furihata, Yukio Nemoto, Hiroyuki Iwasa, Koichi Tan-No, Takeshi Tadano. Effect of Enterococcus faecalis 2001 on colitis and depressive-like behavior in dextran sulfate sodium-treated mice: involvement of the brain–gut axis. J Neuroinflammation. 2019; 16: 201.

I saw the author tested two different doses such as 3, and 30mg/kg EF-2001 as challenge dose, have you been tested what the toxicity level do they find. In another study, they tested 250 mg/kg of C57Bl/6 mice. How about the SD rat model, have you tested the high range of EF-2001 tested in the SD rat model?

Response to Reviewer Comment : We showed no toxicity of EF-2001 in cell experiments. However, we did not conduct a toxicity test of EF-2001 in rats. 

Dosage determination of EF-2001 was calculated based on the dose administered to humans. The adult clinical dosage of EF-2001 is 1.5 g, once a day. After conversion, the clinical dosage for adults is 25 mg/kg/day (adult weight is considered 60 kg). The equivalent dose for mice is 12.3 times that of human adults, 307.5 mg/kg/day probiotics [Reference 1 and 2].

Also, according to previous reports on EF-2001, various oral concentrations of EF-2001 were administered in our mouse experiments. In Fan’s group, 200 mg/kg of EF-2001 was orally administered in mice [Reference 2]. In Takahashi’s group, 250 mg/kg of EF-2001 was used in mice [Reference 3]. However, the results of toxicity tests with oral administration have been recently reported. The 50% lethal dose of EF-2001 with oral administration in mice was estimated to be greater than 5,000 mg/kg body weight/day for both male and female mice [Reference 4]. We did not conduct a toxicity test of EF-2001 in SD rat experiments, but the maximum concentration administered was 30 mg/kg. This concentration is much lower than those used in other groups and is safe based on the results of toxicity test

[Reference 1] Nair, A.; Morsy, M.A.; Jacob, S. Dose translation between laboratory animals and human in preclinical and clinical phases of drug development. Drug Develop. Res. 2018, 79, 373–382.

[Reference 2] Meiqi Fan, Young-Jin Choi, Nishala Erandi Wedamulla, Yujiao Tang, Kwon Il Han, Ji-Young Hwang, Eun-Kyung Kim. Heat-Killed Enterococcus faecalis EF-2001 Attenuate Lipid Accumulation in Diet-Induced Obese (DIO) Mice by Activating AMPK Signaling in Liver. Foods. 2022 Feb; 11(4): 575. 

[Reference 3] Kohei Takahashi, Osamu Nakagawasai, Wataru Nemoto, Takayo Odaira, Wakana Sakuma, Hiroshi Onogi, Hiroaki Nishijima, Ryuji Furihata, Yukio Nemoto, Hiroyuki Iwasa, Koichi Tan-No, Takeshi Tadano. Effect of Enterococcus faecalis 2001 on colitis and depressive-like behavior in dextran sulfate sodium-treated mice: involvement of the brain–gut axis. J Neuroinflammation. 2019; 16: 201. 

[Reference 4] Yeun-Hwa Gu, Takenori Yamasita, Ki-Mun Kang. Subchronic Oral Dose Toxicity Study of Enterococcus Faecalis 2001 (EF 2001) in Mice. Toxicol Res. 2018 Jan; 34(1): 55–63.

In Fig. 4 do you have any histogram data of cell cycle? Did you observe any cell cycle dependent kinase interrupted or not? Have you tested with positive control compared to EF-2001? Do you have any animal data regarding cell proliferation data to demonstrate your hypothesis (Line 347, author mentioned EF-2001 induce inhibition of early cell cycle signals in the MCE process in both?)

Response to Reviewer Comment : We having the cell cycle histogram of data. In the Figure 4, the data show that S phase were increased, and G2/M phase were decreased by EF-2001. Therefore, it is expressed by analogy because it shows the arrest pattern by looking at the population percentage of each phase of the cell cycle. In our experiments, proteins that regulate each phase of the cell cycle (such as cell cycle-dependent kinases) have not been identified. The exact identification and mechanism of cell cycle regulatory proteins in the current HFD model requires further study with reference to the reviewers' comments. We corrected this and added content to the Discussion.

Am wondering they verify the signaling crosstalk if EF-2001 activated in the rat model by confirming gene and protein level expression following EF-2001 and determine functional analysis using siRNA technology or others, what molecular targeting underlying IR-beta signaling could be activated anti-obesity effect on Scheme Fig.8.

Response to Reviewer Comment : We agree with you. EF-2001 confirmed the inhibitory mechanism of IR-beta signaling. However, it has not been confirmed through the same signaling crosstalk experiment using siRNA. In the future, research on signaling crosstalk using siRNA will be confirmed. 

Wondering interaction between EF-2001 and enzymatic activity in lipolysis in SD rat tissue.

Response to Reviewer Comment : Thank you for the comments. Unfortunately, AMPK was not confirmed in adipocyte or adipose tissue. Our experimental model shows inhibition of lipid production in adipogenesis based on control of the sub-signal of the receptor that activates the adipogenic transcription factor. The AMPK signal is a protein marker of lipolysis, which demonstrates the effects of drugs that induce lipid production in vivo or in vitro and activate lipolysis enzymes (such as HSL, ATGL and MGL). Phosphorylation of AMPK mediates mechanisms related to beta oxidation or ATP consumption [Reference 1]. Therefore, a difference in the order of intracellular signaling resulted from treatment with EF-2201 during adipogenic induction.

[Reference 1] Aslam, M.; Ladilov, Y. Emerging Role of cAMP/AMPK Signaling. Cells 2022, 11, 308

Is there any evidence of AMPK signaling alteration along with cytokines /chemokines pattern following EF treatment between SD and HFD rats?

Response to Reviewer Comment : Thank you for your comments. Recently, another notable study has reported the influence of weight control based on the impact of EF-2001 administration on the AMPK pathway in mice with obesity induced by HFD. Fan et al. reported that EF-2001 significantly upregulated the expression of p-AMPK and p-ACC in murine liver and suggested that EF-2001 decreases hepatic lipid accumulation in the DIO model mice through the AMPK pathway and ameliorates liver damage caused by an HFD [Reference 1]. AMPK is a key regulator of energy metabolism in mammalian cells and is crucial to maintaining energy homeostasis The AMPK signal is a protein marker of lipolysis, which demonstrates the effects of drugs that induce lipid production in vivo or in vitro and activate lipolysis enzymes (such as HSL, ATGL and MGL). Phosphorylation of AMPK mediates mechanisms related to beta oxidation or ATP consumption [Reference 2]. Therefore, a difference in the order of intracellular signaling resulted from treatment with EF-2201 during adipogenic induction. However, it is necessary to analyze the signaling mechanisms directly related to lipid metabolism in vivo in adipocytes and tissues and to study the mechanisms of EF-2001 from differentiation induction of adipocytes to lipolysis. For a clear relevance analysis, this aspect will require further study.We described in the Discussion.

[Reference 1] Meiqi Fan, Young-Jin Choi, Nishala Erandi Wedamulla, Yujiao Tang, Kwon Il Han, Ji-Young Hwang, Eun-Kyung Kim. Heat-Killed Enterococcus faecalis EF-2001 Attenuate Lipid Accumulation in Diet-Induced Obese (DIO) Mice by Activating AMPK Signaling in Liver. Foods. 2022 Feb; 11(4): 575. 

[Reference 2Aslam, M.; Ladilov, Y. Emerging Role of cAMP/AMPK Signaling. Cells 2022, 11, 308

Minor issue

Would you provide chemical components with the EF-2001?

Response to Reviewer Comment : As you may know, EF-2001 is composed of dead Enterococcus faecalis and various other components. The specific content composition is not available. 

In fig 1A, there are missing figure legends regarding C. Also, do you have a HFD with EH, EL abdomen needs to be indicated when the rat is sacrificed in detail.

Response to Reviewer Comment : Thanks for your comments. Figure 1C and its content were added. Also, we corrected the notation of Figure 1. In addition, to minimize misunderstanding, the labeling was changed from EL to EF-2001 (3 mg/kg) and from EH to EF-2001 (30 mg/Kg).

To clarify your experimental study, suggest a schematic workflow including treatment points of EL and EH with no subject in the animal study.

Response to Reviewer Comment : We added the schematic workflow in Figure 1a to clearly present the animal experiments and explained the contents.

In Fig. 3 describe GM abbreviation in figure legend

Response to Reviewer Comment : We defined the GM abbreviation in the figure legend.
